# CARdioimaging in Lung Cancer PatiEnts Undergoing Radical RadioTherapy: CARE-RT Trial

**DOI:** 10.3390/diagnostics13101717

**Published:** 2023-05-12

**Authors:** Valerio Nardone, Maria Paola Belfiore, Marco De Chiara, Giuseppina De Marco, Vittorio Patanè, Giovanni Balestrucci, Mauro Buono, Maria Salvarezza, Gaetano Di Guida, Domenico D’Angiolella, Roberta Grassi, Ida D’Onofrio, Giovanni Cimmino, Carminia Maria Della Corte, Antonio Gambardella, Floriana Morgillo, Fortunato Ciardiello, Alfonso Reginelli, Salvatore Cappabianca

**Affiliations:** 1Department of Precision Medicine, University of Campania “L. Vanvitelli”, 80138 Naples, Italy; 2Radiotherapy Unit, Ospedale del Mare, ASL Napoli 1 Centro, 80138 Naples, Italy; 3Department of Translational Medical Science, University of Campania “L. Vanvitelli”, 80138 Naples, Italy

**Keywords:** NSCLC, radiotherapy, thoracic imaging, cardiac MRI, cardiac CT

## Abstract

Background: Non-small-cell lung cancer (NSCLC) is a common, steady growing lung tumour that is often discovered when a surgical approach is forbidden. For locally advanced inoperable NSCLC, the clinical approach consists of a combination of chemotherapy and radiotherapy, eventually followed by adjuvant immunotherapy, a treatment that is useful but may cause several mild and severe adverse effect. Chest radiotherapy, specifically, may affect the heart and coronary artery, impairing heart function and causing pathologic changes in myocardial tissues. The aim of this study is to evaluate the damage coming from these therapies with the aid of cardiac imaging. Methods: This is a single-centre, prospective clinical trial. Patients with NSCLC who are enrolled will undergo computed tomography (CT) and magnetic resonance imaging (MRI) before chemotherapy 3 months, 6 months, and 9–12 months after the treatment. We expect to enrol 30 patients in 2 years. Conclusions: Our clinical trial will be an opportunity not only to highlight the timing and the radiation dose needed for pathological cardiac tissue changes to happen but will also provide useful data to set new follow-up schedules and strategies, keeping in mind that, more often than not, patients affected by NSCLC may present other heart- and lung-related pathological conditions.

## 1. Introduction

Non-small-cell lung cancer (NSCLC) is one of the most prevalent lung tumours and is responsible for a significant proportion of cancer-related deaths and illnesses worldwide [1]. Despite global efforts to increase awareness and early detection, a considerable number of NSCLC patients still receive a diagnosis only at advanced stages of the disease, which negatively affects their prognosis and treatment options. According to recent statistics, around 20% of NSCLC cases are diagnosed at advanced stages [2,3]. The late diagnosis of NSCLC results in a great deal of heterogeneity in terms of prognosis and treatment as patients may be eligible for surgery, chemotherapy, radiotherapy, or a combination of these treatments. Modern therapies have been able to achieve unprecedented survival rates, with increasing numbers of patients experiencing long-term remission [4,5,6,7]. However, this success comes at a cost, with numerous side effects associated with these therapies, including radiation-induced toxicity to the heart and coronary vessels [8,9,10,11]. Patients who undergo radiotherapy may experience a significant reduction in their perceived quality of life due to cardiac side effects. However, there is still a general lack of evidence and no universally approved threshold value for the relationship between radiation exposure and the manifestation of clinical side effects involving the cardiac muscle. This lack of evidence makes it challenging to prevent such side effects [12,13,14,15,16]. It is essential to adopt a clinical approach that focuses on maintaining acceptable quality of life for NSCLC patients undergoing treatment, with particular attention paid to managing the numerous side effects associated with modern therapies.

### Study Rationale

In light of the potential cardiac side effects associated with modern NSCLC therapies, cardiac imaging techniques such as cardiac computed tomography (cardiac-CT) and cardiac magnetic resonance (cardiac-MRI) may provide valuable information that could facilitate the tailored management of patients undergoing treatment. Cardiac MRI is considered the gold standard for evaluating ejection fractions and plays a pivotal role in the assessment of myocardial fibrosis [17,18,19,20,21,22,23,24]. Additionally, T2 sequences can easily detect oedema, which may reveal acute myocardial damage [25]. Cardiac-CT, on the other hand, offers greater spatial resolution, allowing for a better understanding of a patient’s anatomy. Furthermore, it can quantify the volume of non-calcified plaques and the extent of coronary stenosis in both symptomatic and asymptomatic patients [26,27]. The combined use of both these techniques may be useful for detecting early and late changes in cardiac vessels and the ventricular muscle. When a baseline examination acquired before treatment is available for comparison, these techniques become even more useful. Given the heterogeneity of cardiac side effects associated with NSCLC treatment and the lack of validated dose constraints in modern radiotherapy, studies using cardiac imaging techniques may be beneficial for understanding radio-induced cardiac toxicity. This information could lead to new opportunities for managing patients and improving their quality of life during treatment.

## 2. Materials and Methods

This prospective study will span fifteen months and will focus on evaluating the acute and chronic cardiac toxicity associated with thoracic radiotherapy by using advanced imaging techniques such as cardiac-MRI and cardiac-CT in patients with NSCLC. In addition to assessing cardiac toxicity, the study will also consider overall survival, disease-free survival, and locoregional control, as well as cardiac dosimetry parameters. The ultimate goal of this study is to use the data collected to predict the cardiological toxicity associated with adjuvant radiotherapy delivered with modern techniques. Furthermore, the study will seek to define the role of imaging methods in the prevention of cardiac risk in the neoadjuvant setting. It is expected that the results of this study will be beneficial for clinicians and researchers alike, providing valuable insights into the cardiac side effects of NSCLC treatment and potential strategies for mitigating those risks. Along with advanced imaging techniques, the study will also collect blood samples for the analysis of serum biomarkers. This approach will enable researchers to relate any pathological changes observed on the imaging studies with corresponding changes in serum biomarkers, providing a more comprehensive understanding of the mechanisms underlying cardiac toxicity in NSCLC patients undergoing radiotherapy [28,29,30,31].

### 2.1. Primary Endpoint

To evaluate early and late myocardial and coronary damage due to radiotherapy.

### 2.2. Secondary Endpoints

To evaluate progression-free survival (PFS).Local control evaluation.Overall survival evaluation.Cardiac, pulmonary, and oesophageal evaluation.

### 2.3. Inclusion Criteria (Table 1)

The histological diagnosis of non-small-cell lung neoplasm.Patients undergoing radiotherapy treatment with radical intent or with adjuvant intent. Intensity-modulated radiation therapy (IMRT) or the volumetric modulated arc therapy (VMAT) technique.

**Table 1 diagnostics-13-01717-t001:** Inclusion and exclusion criteria in CARE-RT trial.

Inclusion Criteria	Exclusion Criteria
• Histological diagnosis of non-small-cell lung cancer	• Metastasis in more than one pulmonary lobe
• Radiotherapy treatment: IMRT or VMAT	• Previous chest radiotherapy
	• Clinical progrssion during postoperative chemotherapy
	• Recent heart/lung disease

### 2.4. Exclusion Criteria (Table 1)

Documented metastases in a different lobe.Previous chest radiotherapy.Clinical progression during postoperative chemotherapy.Recent (inferior to 6 months) heart disease or severe lung disease.

### 2.5. Study Setting

The present prospective study has the potential to contribute valuable insights and data for various purposes related to NSCLC patients undergoing chemo-radiotherapy.

Firstly, it can shed light on the epidemiology of cardiac function in this specific population, revealing that NSCLC patients often fall under certain demographic categories such as being males, current smokers, and above 60 years old. The findings of this study can help with developing future prevention strategies for this vulnerable group.

Secondly, the study can provide a basis for the early post-treatment evaluation of the effect of chemo-radiotherapy on cardiac functionality. This can be crucial in developing future cardiac-sparing strategies, ensuring that the treatment does not cause adverse effects on the cardiac health of the patient.

Thirdly, the study can help in the late post-treatment evaluation of cardiac functionality after chemo-radiotherapy, which can be important in tailoring the follow-up of cancer survivors and preventing cardiac events in this population. Regular follow-up can help detect any adverse effects on cardiac function and implement appropriate measures to prevent further deterioration.

The delta analysis, which takes into consideration the differences between late and early post-treatment imaging and baseline imaging, will be pivotal in quantifying the effects of chemo-radiotherapy in patients with NSCLC. This analysis can help develop future strategies to prevent cardiovascular side effects and establish an effective follow-up plan for these patients.

A complete evaluation of the serum biomarkers’ ability to assess the development of cardiac functionality impairment is required. This task can be performed through the evaluation of changes in endothelial progenitor cells (EPCs) levels before and after treatment. In fact, EPCs represent an integral component of cardiovascular homeostasis, participating in the process of endothelial repair after vascular injury. Reduced levels of EPCs were associated with a significant increased risk of cardiovascular events, all-cause death, and onset/progression of microangiopathy [32]. The assessment of circulating levels of EPCs will be performed on fresh blood samples. Peripheral blood cells will be analyzed for the expression of surface antigens CD34+, KDR+, and CD133+ by direct flow cytometry. Quantitative analysis will be performed on a BD FACSCalibur cytometer (Becton-Dickinson), and 1,000,000 cells will be acquired in each sample. With this task, we will evaluate whether radiotherapy may impair endothelial function and reduce EPCs levels, increasing the risk of cardiovascular events.

A special database has been developed to list the data obtained through our study. The variables that will be collected are summarized below:Data scope;Personal data such as gender, age, and date of recruitment;Clinical history with special care for comorbidity and smoking habit;General status assessment Eastern cooperative oncology group (ECOG) performance status;TNM, histology, and molecular diagnostics;Radiotherapy treatment, possible concomitant or sequential chemotherapy;Response to therapy according to RECIST criteriaRadiotherapy parameters (technique, total dose, dose per fraction, volumes, start/end date, V5, V30, MeanDose, DMAX Cardiac Dosimetry, MeanDose Pulmonary Dosimetry, V20, V30, MeanDose Esophageal Dosimetry, V50, and V55);CT parameters (calcium score, thrombosis, and calcified plaques)Cardiac-MRI parameters (tissue evaluation, ventricular functional analysis, valve condition, and atrial dimension);Acute and late toxicity according to common terminology criteria for adverse events (CTCAE) 4.0 scales;Clinical outcomes overall survival (alive, deceased);Progression-free survival (progression yes/no, date of progression);Local control;Symptom control.

This research will measure two important outcomes: overall survival and progression-free survival. Overall survival refers to the amount of time from surgery to death, while progression-free survival refers to the amount of time from surgery to the point when the disease starts to progress. To analyze these outcomes, we will use the Kaplan–Meier method. This is a statistical method commonly used in medical research to estimate the probability of survival over time. By applying this method to the data collected from the study participants, the researchers will be able to determine the overall and progression-free survival rates for the group. In addition to analyzing survival rates, the study will also assess the acute and late toxicity of the radiotherapy treatment. Acute toxicity refers to any adverse effects that occur during or immediately after treatment, while late toxicity refers to adverse effects that occur at least 6 months after the end of treatment. The physicians involved will report the crude incidence rates of these toxicities, which will provide important information about the safety and tolerability of the treatment. To evaluate the impact of the treatment on cardiac function and coronary involvement, CT-scan and cardiac-MRI imaging will be performed. This will allow one to assess the function of the heart and its blood vessels before, during, and after treatment. The researchers will evaluate these factors at multiple time points: before treatment, 6 months from the beginning of treatment, 9 months from the beginning of treatment, and after 12–15 months from the end of treatment. Finally, quantitative methods to evaluate any early and late effects of the treatment on cardiac functionality will be used. This will provide additional insights into the potential long-term impact of the treatment on the heart and its function. Overall, this study aims to provide valuable information about the efficacy and safety of radiotherapy treatment for a specific tumour type, as well as its potential impact on cardiac function and coronary involvement. By analyzing multiple outcomes and using advanced imaging techniques, we believe we will gain a comprehensive understanding of the effects of targeted radiotherapy.

### 2.6. CT Setting

The multi-slice computed tomography (MSCT) scan is a non-invasive medical imaging technique that provides detailed images of the human body. In this particular case, a 64-slice MSCT scanner, the Revolution EVO (GE Healthcare in Little Chalfont, Buckinghamshire, UK), will be used to scan patients. To evaluate the coronary arteries, ECG retrospective gating will be applied, which synchronizes the scanning process with the patient’s heartbeat. This ensures that the images captured are clear and accurate. Immediately after the ECG-gated scan, late-portal and delayed-phase images will be acquired. These images will provide additional information about the structures and tissues in the body. To enhance the images, an iodinated contrast agent called Xenetix (iobitridol, 350 mg/mL, manufactured by Guerbet in Villepinte, France) will be used. This contrast agent helps to highlight blood vessels and other structures in the body, making it easier to identify any abnormalities or lesions. During the scanning process, the patient will lie on a table that moves through the scanner. The scan will be performed in a cranio-caudal direction, which means that the scanner will capture images from the head to the feet. The collimation will be set to 64 × 0.6 mm, which means that the scanner will capture 64 images per rotation, with each image covering a slice of the body that is 0.6 mm thick. The pitch will be set between 0.2 and 0.5, which determines the amount of overlap between the images captured. A pitch of 0.2 means that there will be a 20% overlap between images, while a pitch of 0.5 means that there will be a 50% overlap. The gantry rotation time will be set to 3.30 s, which means that the scanner will rotate around the patient in 3.30 s. The pipe current will be set to 330 mA, which is the amount of electrical current that passes through the X-ray tube during the scan. This determines the amount of radiation that the patient is exposed to. The esp-dependent pipe power will be set to 120 kV for patients weighing more than 85 kg, and 100 kV for patients weighing less than 85 kg. This determines the voltage that is applied to the X-ray tube during the scan, which also affects the amount of radiation that the patient is exposed to. To ensure that the contrast agent is delivered to the correct area, a contrast smart prep ROI will be positioned on the aortic arch, and a threshold of 100 HU will be set. This will ensure that the contrast agent is delivered to the blood vessels in the correct location, enhancing the images of the coronary arteries and other structures in the body. Overall, the 64-slice MSCT scan with ECG gating, delayed-phase images, and the iodinated contrast agent will provide detailed images of the patient’s coronary arteries and other structures in the body, enabling accurate diagnosis and treatment (Table 2).

### 2.7. MR Setting

Magnetic resonance imaging (MRI) is a non-invasive medical imaging technique that uses a magnetic field and radio waves to create detailed images of the body. In this case, a 1.5 T scanner, the SIGNA Voyager (GE Healthcare in Little Chalfont, Buckinghamshire, UK), will be used to perform the MRI scan. To enhance the images, a contrast agent called Gadovist (gadobutrol, 1.0 mmol/mL, manufactured by Bayer AB in Leverkusen, Germany) will be injected intravenously into the patient (Table 2). This contrast agent helps to highlight areas of interest in the body, making it easier to identify any abnormalities or lesions. The acquisition protocol will require balanced steady-state free precession sequences oriented along the main cardiac axes and short-time inversion recovery sequences with the same orientation. This will help to capture clear images of the heart and surrounding structures, allowing for accurate diagnosis and treatment. T1 weighted volumetric pulse sequences will be applied 10 min after the injection of contrast agent. This will help to evaluate cardiac function as well as the presence of myocardial oedema, which is the accumulation of fluid in the heart muscle that can be a sign of various heart conditions. Perfusion sequences will also be acquired, which will provide additional information about blood flow to the heart and any areas of scar tissue or damage. Imaging analysis will be performed using a dedicated software called CVi42 (manufactured by Circle Cardiovascular Imaging in Calgary, AB, Canada). This software is designed specifically for cardiac MRI analysis and allows for the accurate measurement of cardiac function, blood flow, and tissue characteristics. Overall, the MRI scan with a contrast agent and specialized imaging sequences will provide detailed images of the heart and surrounding structures, allowing for the accurate diagnosis and treatment of various heart conditions. The use of dedicated software for analysis will also ensure the accurate interpretation of the images and facilitate effective communication between healthcare professionals.

## 3. Discussion

The importance of investigating cardiologic follow-up in NSCLC patients undergoing chemo-radiotherapy (+/− immunotherapy) cannot be overstated. This is because such treatment protocols have been associated with cardiotoxicity, which can lead to serious cardiac complications if left unchecked. The early and late detection of cardiotoxicity is therefore crucial in ensuring that patients receive steady and effective treatment. One potential strategy for detecting cardiotoxicity is through the use of cardiac imaging techniques such as echocardiography, CT scan, and MRI. However, it is important to note that echocardiography, despite being widely available and easy to perform, has been found to have poor reproducibility in calculating LV function, global longitudinal strain (GSL), and speckle tracking [33]. As a result, researchers are turning to other imaging techniques such as the CT scan and MRI for the more accurate assessment of cardiac function. 

CT anatomic imaging is advantageous for detecting non-obstructive coronary artery disease CAD early in the disease process before ischemia develops, allowing for the earlier use of preventive medications. Coronary computed tomographic angiography (CCTA) can noninvasively detect and quantify non-calcified plaque and coronary stenosis and has a powerful negative predictive value when ruling out CAD, with lower radiation exposure than other modalities. On the other hand, coronary artery calcification (CAC) scans are able to evaluate calcified plaque and have even lower radiation exposure. CAC scans can identify coronary atherosclerosis in patients with cancer before RT treatment and might predict cardiovascular outcomes. Cardio CT is important for the study of cardiotoxicity and can be combined with clinical data using machine learning models for individualized risk prediction [34,35,36,37,38].

The use of cardiac MRI has become increasingly popular in clinical practice due to its ability to provide valuable information about anatomical and tissue changes that occur in the heart during pathological conditions [39]. Techniques such as late gadolinium enhancement have become more established, while newer techniques such as T1, T2, and T2 mapping have seen a significant rise in use and development [40]. While there are challenges regarding the practical utilization, accuracy, and reproducibility of cardiac MRI, it has shown higher accuracy and reproducibility than cardiac ultrasound [41]. Cardiac MRI is particularly useful in cases of new cardiac symptoms such as arrhythmias, myocarditis, or reduced ventricular function and can provide valuable diagnostic information for uncommon adverse effects resulting from new anti-tumoral drugs such as immune checkpoint inhibitors [42]. Studies have shown that patients with ICI myocarditis have significantly elevated T1 and T2 values, which have been linked to more pronounced symptoms and lower cardiac function [43]. Additionally, T1 elevation in acute myocarditis may be related to oedema and has independent prognostic value for the development of major adverse cardiac events [44].

In this protocol, researchers will investigate the potential adverse effects of chest radiotherapy on the heart using the CT scan and MRI (Figure 1 and Figure 2). They will evaluate the coronary vessels and myocardial tissue to determine if and under what circumstances radiation caused heart damage. This information could then be used to develop new follow-up schedules and strategies to ensure that patients are effectively monitored for cardiotoxicity. Other authors have also been investigating the cardiotoxicity of RT using different techniques and protocols. However, the use of MRI is recognized as having its disadvantages, including high cost, long image-acquisition time, and limited availability. Until these challenges can be addressed, its use in cardio-oncologic populations will likely remain limited to those in whom echocardiographic assessments are of poor quality or inconclusive [45,46,47,48,49,50,51,52,53,54].

Therefore, the study discussed above could be pivotal in understanding the added value of MRI imaging investigations in NSCLC patients undergoing RT. It is also noteworthy that the clinical implementation of MRI-Linac will shed light on the usefulness of MRI for organs at risk, including the heart [55,56,57,58]. This could lead to more effective strategies for detecting and managing cardiotoxicity in cancer patients, ultimately improving outcomes and quality of life.

## Figures and Tables

**Figure 1 diagnostics-13-01717-f001:**
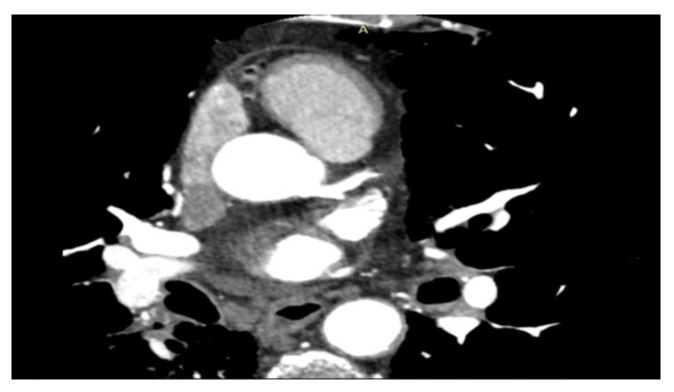
Coronary CT scan.

**Figure 2 diagnostics-13-01717-f002:**
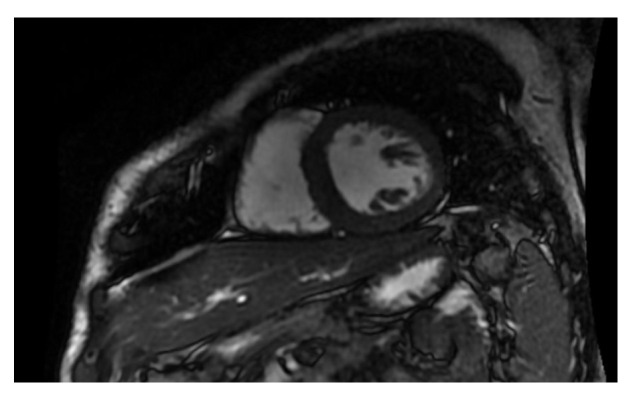
Cardiac MRI, ventricular short axis.

**Table 2 diagnostics-13-01717-t002:** CT and MRI settings used in CARE-RT trial.

CT Settings	MR Settings
• Collimation 64 × 0.6 mm	• 1.5 Tesla Magnet
• Pitch 0.2 to 0.5	• Slice thickness 8 mm
• Gantry rotation time 3.30 s	• Slice spacing 2 mm
• Pipe current 330 mA	• T1 FS sequences 10 min after the contrast agent injection
• Pipe power 120 kV (patient > 85 kg) or 100 kV (patient < 85 Kg)	• Prospective cardiac synchronization
• ROI threshold 100 HU	
• Slice thickness 1.0 mm	

## Data Availability

The data presented in this study are available on request from the corresponding author. The data are not publicly available due to privacy and ethical restrictions.

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
