# Peer review of "CARdioimaging in Lung Cancer PatiEnts Undergoing Radical RadioTherapy: CARE-RT Trial"

_diagnostics, 2023, doi:10.3390/diagnostics13101717_

Round 1

Reviewer 1 Report

In this manuscript, the authors describe a protocol to follow up on the effects of chemo- and radio-therapy on patients that suffered from non-small cell lung cancer.

The topic is interesting and important. The manuscript is also well-written. I only have some formal comments.

Detailed comments and suggestions:

  1. Figures 1 and 2 are neither mentioned nor described in the text. Their presence is justified only if the authors want to discuss something related to coronary CT or cardiac MRI. Otherwise, they can be removed.

  2. Please report the first time that they are introduced the extended names of the acronyms. I have highlighted some cases where this is necessary.

  3. Please see the attached file for some highlighted typos.

Author Response

Thank you for taking the time to review our manuscript.
We appreciate your valuable input and have carefully considered your suggestions. Specifically, we have made the necessary revisions to address the highlighted typos as per your feedback.

Furthermore, we believe that coronary-CT and cardiac-MRI, being less common diagnostic methods,  may not be familiar to all readers. Therefore, we have included images 1 and 2 to assist those who may not be familiar with these methods, thereby aiding in better comprehension of the topic.

Thank you again for your time and effort in reviewing our manuscript.

Best regards Marco De Chiara

Reviewer 2 Report

# General Comments

Thank you for the opportunity to review this manuscript. The authors established the protocol of CARE-RT to evaluate the damage coming from a combination of chemotherapy, radiotherapy, and immunotherapy with the aid of cardiac imaging. Recently, the value of cardiac toxicity has been more and more concerned, and the results of this study would be highly noted due to its novelty.

# Minor point

How did the authors calculate the required sample size? The clarification of this would improve the protocol.

Author Response

Dear revisor,

We would like to bring to your attention that the study in question is a preliminary investigation aimed at helping us define the timing of subsequent protocols. We want to clarify that the sample size was calculated by taking into account the incidence of pathology (stage 3, which is usually treated with 
chemotherapy and/or radiotherapy) in our Institute, along with the deadline of 2 years.

We appreciate your time and effort in reviewing our study,  and we hope that this additional information provides a better understanding of the context in which the research was conducted. If you have any further questions or concerns, please do not hesitate to contact us.

Thank you for your attention.

Best regards, Marco De Chiara